# Practical Issues in Early Switching from Intravenous to Oral Antibiotic Therapy in Children with Uncomplicated Acute Hematogenous Osteomyelitis: Results from an Italian Survey

**DOI:** 10.3390/ijerph16193557

**Published:** 2019-09-23

**Authors:** Elena Chiappini, Elena Serrano, Luisa Galli, Alberto Villani, Andrzej Krzysztofiak

**Affiliations:** 1Paediatric Infectious Disease Unit, Meyer Children’s University Hospital, Department of Health Sciences, University of Florence, 50100 Lorence, Italy; elenaserrano0902@gmail.com (E.S.); luisa.galli@unifi.it (L.G.); 2Pediatric and Infectious Diseases Unit, Academic Department, Bambino Gesù Pediatric Hospital, 00146 Rome, Italy; alberto.villani@opbg.net (A.V.); andrzej.krzysztofiak@opbg.net (A.K.)

**Keywords:** acute hematogenous osteomyelitis, children, antibiotic

## Abstract

*Background*: The European Society of Pediatric Infectious Diseases (ESPID) guidelines for acute hematogenous osteomyelitis (AHOM) have been published recently. In uncomplicated cases, an early (2–4 days) switch to oral empirical therapy, preferentially with flucloxacillin, is recommended in low methicillin-resistant *Staphylococcus aureus* settings. We conducted a survey with the aim of evaluating the behaviors of Italian pediatricians at this regard. *Methods*: An open-ended questionnaire investigating the empiric therapy adopted in uncomplicated AHOM children according to age was sent by email to 31 Italian pediatric clinics taking care of children with infectious diseases, and results were analyzed. *Results*: The preferred intravenous (IV) regimen was a penicillin plus an aminoglycoside (*n* = 10; 32.3%) in children aged <3 months, and a combination of a third-generation cephalosporin plus oxacillin (*n* = 7; 22.6%), or oxacillin alone (*n* = 6; 19.4%) in those ≥3 months. In every age class, amoxicillin-clavulanate was the first-choice oral antibiotic. Other antibiotics largely used orally included clindamycin, rifampicin, and trimethoprim/sulfamethoxazole. Flucloxacillin was never prescribed. Only 3 centers switched to oral therapy within 7 days in children ≥3 months of age. The most commonly reported reason influencing the time to switch to oral therapy concerned caregivers’ adherence to oral therapy. *Conclusion*: Adherence to guidelines was poor, and early transition to oral therapy in the clinical practice was rarely adopted. Given the large use of potentially effective, but poorly studied, oral antibiotics such as amoxicillin/clavulanate, trimethoprim/sulfamethoxazole, and rifampicin, our data may stimulate further studies of this regard.

## 1. Introduction

Acute hematogenous osteomyelitis (AHOM) is the most common bone infection in infants and children [1]. Since the pathogen identification is not obtained in the majority of cases, empirical therapy, based on local epidemiological data, is fundamental [1]. Several previous surveys demonstrated heterogeneous behaviors among pediatricians in France and Spain [2,3]. A recent European Society of Pediatric Infectious Diseases (ESPID) guideline has been released with the aim to uniform AHOM management among European countries [4]. Attention is paid to the optimal empiric antibiotic regimen in different scenarios, on the basis of the child’s age, and local prevalence of methicillin-resistant *Staphylococcus aureus* (MRSA) [1,4,5,6]. The document underlines high heterogeneity regarding the empirical antibiotic treatment adopted among several European countries, based on the panel’s expertise. However, information regarding the therapeutic attitude among Italian physicians is not provided. Moreover, several aspects are still under debate, including the choice of the optimal oral antibiotic therapy and the most appropriate time to switch from intravenous (IV) to oral (PO) regimens. Issues have also been raised regarding caregivers’ adherence to PO therapy when the chosen antibiotic should be administered 4 times a day and/or has poor palatability (i.e., flucloxacillin), or poor literature data are available on its efficacy in pediatric AHOM (i.e., amoxicillin/clavulanate, trimethoprim/sulfamethoxazole (TMP/SMX), rifampicin); or tolerability may be an important concern (i.e., risk of severe rash or *Clostridium difficile* diarrhea using clindamycin); or the achievement and maintenance of adequate drug concentrations in the site of infection may be impaired by the low penetration of the drug into the bone tissue (i.e., cephalexin, cefuroxime). Therefore, it is difficult to recommend a univocal approach, and tailored therapy might be necessary. Moreover, an early (2–4 days) switch to oral therapy is recommended [4], unless risk factors are present, when clinical and laboratory improvement has been observed. This recommendation takes into account several study results demonstrating similar success rates, but lower rates of vascular catheter-related complications, in the group of children receiving short IV therapy versus those treated with prolonged IV courses [7,8,9].

However, some authors described an attitude among pediatricians to prolong the “short” IV therapy to about 7 days in the real-world setting [10], and some experts would suggest a more flexible approach, since the optimal duration of IV antibiotics has not yet been rigorously defined [6].

Considering the complexity of these problems we conducted a national survey in order to evaluate the behaviors of Italian pediatricians in the management of infants and children with uncomplicated AHOM and their adherence to the ESPID recommendations.

## 2. Methods

The aim of this study was to evaluate the empirical therapeutic management of uncomplicated AHOM in infants and children adopted in several Italian pediatric clinics distributed throughout the nation (Table 1). The study was carried out by sending, by email, an open-ended questionnaire developed by an expert panel of the Italian Society of Pediatric Infectious Disease (ISPID) and Italian Society of Pediatrics (ISP), between November 2018 and January 2019. The questionnaire was sent to the Italian pediatric clinics, taking care of infants and children with pediatric infectious diseases, included ISP/ISPID mailing lists.

The questionnaire investigated the main antibiotic drugs used for empiric therapy of uncomplicated osteomyelitis in infants and children aged both <3 and ≥3 months. The questions concerned number of children with AHOM managed annually, the antibiotic used as a first choice in IV and PO therapy, the duration of IV and IV plus PO therapy, and the main factors guiding the switch from IV to PO therapy (Appendix A).

## 3. Results

Thirty-one centers out of 45 (68.9%) from 19/20 Italian regions were included in the study. Overall, they take care of about 255 AHOM infants and children every year (Table 1).

### 3.1. First-Choice Antibiotic Therapy for Empiric Therapy in Children Aged < 3 Months

First-choice empirical IV therapy was the combination of two antibiotics for 27/31 centers (87.1%) and one antibiotic for 4/31 centers (12.9%). The preferred antibiotic regimens were the combination of penicillin plus aminoglycoside (32.3%, *n* = 10), or a third-generation cephalosporin plus oxacillin (22.6%, *n* = 7), or a third-generation cephalosporin plus a glycopeptide (19.4%, *n* = 6) (Table 1).

First-choice empirical PO therapy was one antibiotic for 26/31 centers (83.9%) and two antibiotics for 3/31 centers (9.7%). Amoxicillin-clavulanate was the first choice preferred by 20/31 centers (64.5%). Two out of 31 centers declared to never shift to oral therapy (6.5%). IV duration was 15–28 days in most 19/31 centers (61.3%), and total duration of IV plus PO therapy was 28–42 days in 24/31 (77.4%) centers (Figure 1).

### 3.2. First-Choice Antibiotic Therapy for Empiric Therapy in Children Aged ≥3 Months

First-choice IV therapy was the combination of two antibiotics for 16/31 centers (51.6%) and one antibiotic for 15/31 centers (48.4%). The preferred antibiotic regimens were the combination of a third-generation cephalosporin plus oxacillin (22.6%, *n* = 7), or penicillin (19.4%, *n* = 6), or a third-generation cephalosporin alone (16.1%, *n* = 5), or a third-generation cephalosporin plus a glycopeptide (16.1%, *n* = 5) (Table 1).

First-choice PO therapy was one antibiotic for 28/31 centers (90.3%), and the association of two antibiotics for 3/31 centers (9.7%). Amoxicillin-clavulanate was the preferred molecule by 21/31 centers (67.7%) (Table 1). The duration of IV therapy was 7–14 days in most centers (14/31; 45.2%), while the most commonly observed total duration (IV plus PO) was 29–42 days (15/31; 48.4%) (Figure 1).

Factors considered for the switch from IV to oral therapy are reported in Figure 2. The most commonly reported reason affecting the time to switch to oral concerned caregivers’ adherence to oral therapy.

## 4. Discussion

The present survey allows to evaluate the empirical therapeutic choices for the treatment of uncomplicated AHOM in infants and children adopted in Italian clinics throughout the country. The Results highlight that behaviors are not homogenous and discrepancies exist between clinical practice and ESPID recommendations. The ESPID guidelines recommended that empirical IV therapy should be based on the patient’s age and the country’s prevalence of MRSA, that is below 10% in Italy [1].

Our study shows that in children younger than 3 months most of the centers followed the ESPID guidelines recommendations since the first-choice empirical IV therapy was a combination of penicillin plus an aminoglycoside and total duration of IV plus PO therapy was 4–6 weeks. However, in older children, first-choice empirical IV therapy was performed with a combination of a third-generation cephalosporin plus oxacillin by the majority of centers, differently from the ESPID recommendations (one antibiotic between cefazolin or cefuroxime or anti-staphylococcal penicillin—only if the age is over 5 years). We may speculate that clinicians preferred to prescribe a third-generation cephalosporin in order to reach a larger spectrum. Indeed, a recent multicenter retrospective Italian study showed a high heterogeneity of bacteria isolated in infants and children with AHOM [11]. Moreover, in children ≥3 months of age, the total duration of the treatment was longer in most Italian centers (4–6 weeks) compared to the ESPID recommendation (3–4 weeks). Other discrepancies from the European recommendations regard to duration of IV therapy. Only a few Italian centers carried out a short IV therapy (<7 days) and then switched to oral therapy. It is interesting to notice that the main reason influencing the transition to oral therapy was a concern regarding caregivers’ adherence. Oral 1st–2nd generation cephalosporins and clindamycin were rarely used by Italian pediatricians. Considering that first-generation cephalosporins, anti-staphylococcal penicillins, and clindamycin are the most studied antibiotics in children with AHOM and are recommended by the EPSID guidelines, their use should be solicited in the Italian setting.

Despite oral flucloxacillin is recommended, this molecule is not used in any Italian center.

Practical issues that obstacle the use of flucloxacillin are its low palatability and the 4 times a day administration. Several authors have underlined the fact that adherence to oral flucloxacillin in children is difficult to achieved [12]. This issue is even more pronounced in Italy since flucloxacillin oral liquid is not available and only 1 g tablets are marketed [13].

In most centers first-choice oral therapy was amoxicillin-clavulanate. This regimen may be a good alternative, since it has an appropriate activity for methicillin-susceptible *Staphylococcus aureus*, and pharmacokinetics/pharmacodynamics profile may be suitable for the treatment of AHOM. On the other hand, few published data are available and this regimen has a higher reported rate of adverse events when compared with other narrow-spectrum molecules [14,15,16]. However, it should be noticed that even if a higher rate of adverse events of amoxicillin-clavulanate has been underlined in the ESPID guidelines, these events are usually not severe and transitory [17].

Another possible option for the oral therapy, adopted by some Italian centers, is TMP/SMX, especially in younger children in whom *S. aureus* and *K. kingae* are frequently reported. Indeed, the ESPID guidelines recommend to consider TMP/SMX as an alternative treatment, even though literature data are limited.

Finally, some centers used rifampicin (in combination with other oral molecules, since resistance develops readily in monotherapy), considering its good anti-staphylococcal activity and its high intra-osteoblastic penetration [18].

Our survey has several limitations. Self-reported behaviors can be misleading since some participants might not complete the survey as carefully as they would do in real settings. Moreover, our results may not be generalized to all the pediatric clinics in Italy. However, it should be considered that AHOM is a rare disease and affected infants and children are usually referred to centers with a dedicated pediatric infectious disease center. In our survey all the major Italian pediatric clinics (i.e., Regional Referring Centers for other infectious diseases including HIV infection or tuberculosis) were all included, allowing to reach a declared global number of 255 AHOM children followed per year.

In conclusion, our data point out that Italian hospitalists have incompletely integrated current evidence on AHOM antimicrobial management into their clinical practice and targeted educational interventions are needed at this regard. Moreover, our results raise some concerns regarding the feasibility of an early transition to PO antibiotic therapy in the clinical practice. Finally, given the large use of potentially effective, but poorly studied, oral antibiotics such as amoxicillin/clavulanate, TMP/SMX, and rifampicin, our data may stimulate further studies investigating the role of these antibiotics in treating pediatric AHOM.

## Figures and Tables

**Figure 1 ijerph-16-03557-f001:**
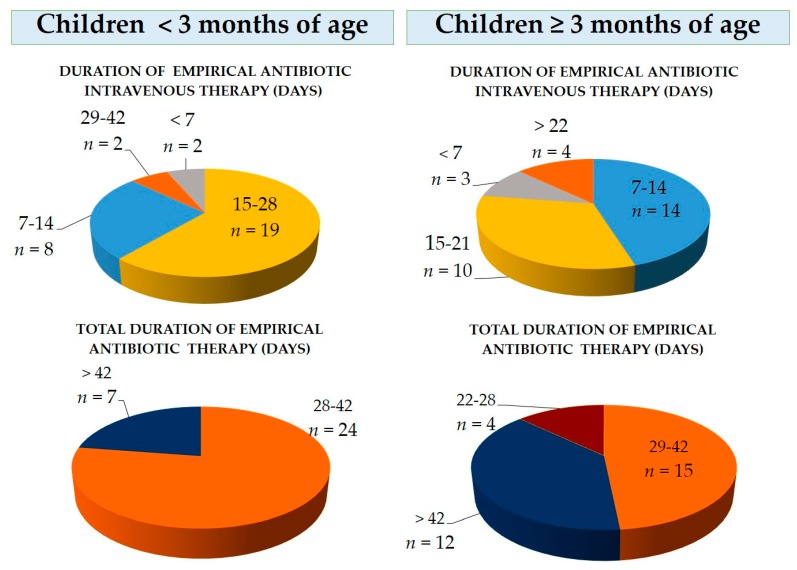
Duration of intravenous and total antibiotic therapy in 31 Italian pediatric clinics.

**Figure 2 ijerph-16-03557-f002:**
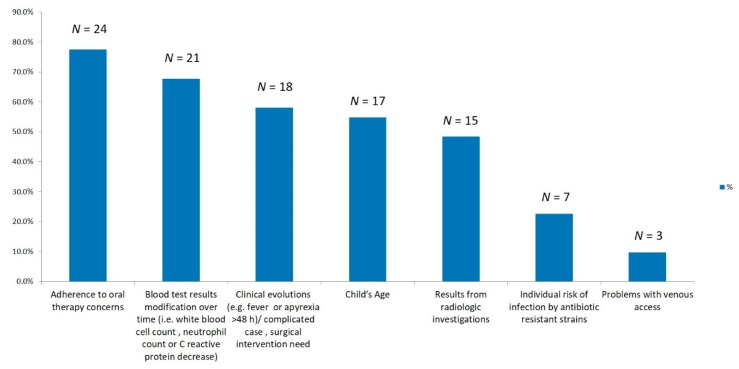
Factors influencing the switch from intravenous therapy to oral therapy in 31 Italian pediatric clinics.

**Table 1 ijerph-16-03557-t001:** Summary of answers to the survey by the 31 participating centers.

Characteristics of the Included Centers and Used Antibiotic Therapy	Total
Geographic location of the Center	
Northern Italy	18
Central Italy	6
Southern Italy	6
Island Regions	1
Number of children with acute hematogenous osteomyelitis (AHOM) followed per year by the Center	
<10 children	21
≥10 children	10
First choice empirical intravenous antibiotics in children under 3 months of age	
Penicillin (Oxacillin, Ampicillin-Sulbactam, Amoxicillin-Clavulanate)	3/31 (9.7%)
3rd gen Cephalosporin	1/31 (3.2%)
3rd gen Cephalosporin + Aminoglycoside (Gentamicin, Netilmicin)	3/31 (9.7%)
3rd gen Cephalosporin + Rifampicin	1/31 (3.2%)
3rd gen Cephalosporin + Oxacillin	7/31 (22.6%)
3rd gen Cephalosporin + Glycopeptide	6/31 (19.4%)
Penicillin (Ampicillin, Oxacillin) + Aminoglycoside (Gentamicin, Netilmicin)	10/31 (32.3 %)
First choice empirical oral antibiotics in children under 3 months	
Never shift to oral therapy	2/31 (6.5%)
Amoxicillin-Clavulanate	20/31 (64.5%)
3rd gen Cephalosporin	3/31 (9.7%)
Clindamycin	3/31 (9.7%) 3
2nd gen Cephalosporin + Rifampicin	1/31 (3.2%)
Amoxicillin-Clavulanate + Rifampicin	1/31 (3.2%)
Amoxicillin-Clavulanate + Clindamycin	1/31 (3.2%)
First choice empirical intravenous antibiotics in children ≥3 months of age	
Penicillin (Oxacillin)	6/31 (19.4%)
1st gen Cephalosporin	3/31 (9.7%)
3rd gen Cephalosporin	5/31 (16.1%)
Clindamycin	1/31 (3.2%)
3rd gen Cephalosporin + Glycopeptide	5/31 (16.1%)
3rd gen Cephalosporin + Oxacillin	7/31 (22.6%)
3rd gen Cephalosporin + Clindamycin	3/31 (9.7%)
3rd gen Cephalosporin + Rifampicin	1/31 (3.2%)
First choice empirical oral antibiotics in children ≥3 months of age	
Amoxicillin-Clavulanate	21/31 (67.7%)
1st gen Cephalosporin	1/31 (3.2%)
3rd gen Cephalosporin	1/31 (3.2%)
Clindamycin	2/31 (6.5%)
Clarithromycin	1/31 (3.2%)
Amoxicillin-Clavulanate + Rifampicin	4/31 (12.9%)
Cotrimoxazole + Rifampicin	1/31 (3.2%)

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
