# Peer review of "Practical Issues in Early Switching from Intravenous to Oral Antibiotic Therapy in Children with Uncomplicated Acute Hematogenous Osteomyelitis: Results from an Italian Survey"

_ijerph, 2019, doi:10.3390/ijerph16193557_

Round 1

Reviewer 1 Report

In general, I would prefer to replace "children" with "infants and children". Would be useful to know the total number of centres that were canvassed in order to know the percentage that responded. Was it 100% that responded? There is a suggestion about this in the Discussion section (lines 170 to 172), so at least this could be made clearer and brought from the Discussion section to the Methods section. Is it known how many centres included in the survey routinely perform blood culture? I assume that as the survey involves haematogenous osteomyelitis, that 100% perform blood culture. Is it known how many routinely perform biopsy and/or tissue culture? I guess that as this question was not included in the questionnaire (lines 186 to 202) then the answer might not be available. It would be good to have a definition up-front of "adherence" - because I am not certain if this is adherence to guidelines or adherence by parents to the administration of oral antibiotics.

Line 48: change "Few" to "Several"

Line 50: remove "the" before "AHOM"

Line 53: change "a highly heterogeneity" to "high heterogeneity"

Line 59: change "molecule" to "antibiotic"

Line 60: change "AHOM children" to "paediatric AHOM"

Line 66: remove "and" and replace with a comma

Line 76: insert "their" before "adherence"

Line 87: replace "followed" with "managed"

Line 90: typo "participating"

Line 92: change "Globally they took care of" to "Overall they take care of"

Lines 134 to 136: did this retrospective study include some or all of the centres in the present survey?

Lines 146-147: This is somewhat of an over-statement about the frequency of Kingella kingae (and no reference provided). Unless this is a particular problem in Italy, it would be best to remove this sentence.

Line 179: again change "molecules" to "antibiotics"

In general, this is a useful document for guiding future strategies in Italy and the graphics are particularly helpful. Suggestions as to the actual types of studies which should be performed would be helpful to investigators.

Author Response

Point by point reply to the referee’s comments

Thanks for your suggestions that have been all included in the present version of our manuscript. Please find a point-by-point reply to each comment below

In general, I would prefer to replace "children" with "infants and children".

REPLY: the word “children" has been replace “infants and children” where appropriate

Would be useful to know the total number of centres that were canvassed in order to know the percentage that responded. Was it 100% that responded? There is a suggestion about this in the Discussion section (lines 170 to 172), so at least this could be made clearer and brought from the Discussion section to the Methods section.

REPLY. 31 out of 45 (68.9%) centers replied. We specified this information in the text.

Is it known how many centres included in the survey routinely perform blood culture? I assume that as the survey involves haematogenous osteomyelitis, that 100% perform blood culture. Is it known how many routinely perform biopsy and/or tissue culture? I guess that as this question was not included in the questionnaire (lines 186 to 202) then the answer might not be available.

REPLY. This question was not included but we believe that all the Centers performed blood culture.

It would be good to have a definition up-front of "adherence" - because I am not certain if this is adherence to guidelines or adherence by parents to the administration of oral antibiotics.

REPLY. Actually sometime the word adherence refers to guidelines and sometimes to parents. We changed the sentences that may be misleading. In particular the following sentences has been modified

Page 1. Lines 38-39 . The most commonly reported reason influencing the time to switch to oral therapy concerned caregivers’ adherence to oral therapy. Page 2. Lines 58-59 . Issues have also been raised regarding caregivers’ adherence to PO therapy when the chosen molecule should be administered 4 times a day and/or has poor Page 5. Lines 119-121. Factors considered for the switch from IV to oral therapy are reported in Figure 2. The most commonly reported reason affecting the time to switch to oral concerned caregivers adherence to oral therapy. Page 5. Lines 141- 143. It is interesting to notice that the main reason influencing the transition to oral therapy was a concern regarding caregivers’ adherence.

Line 48: change "Few" to "Several"

REPLY: the word "Few" has been replace "Several"

Line 50: remove "the" before "AHOM"

REPLY: the word “the” has been remove

Line 53: change "a highly heterogeneity" to "high heterogeneity"

REPLY: the words "a highly heterogeneity" have been replace "high heterogeneity"

Line 59: change "molecule" to "antibiotic"

REPLY: the word "molecule" has been replace "antibiotic”

Line 60: change "AHOM children" to "paediatric AHOM"

REPLY: the words “AHOM children" have been replace "paediatric AHOM"

Line 66: remove "and" and replace with a comma

REPLY: the word “and” has been replace with a comma

Line 76: insert "their" before "adherence"

REPLY: the word “their” has been insert before “adherence”

Line 87: replace "followed" with "managed"

REPLY: the word "followed" has been replace "managed"

Line 90: typo "participating"

REPLY: the word has been correct

Line 92: change "Globally they took care of" to "Overall they take care of"

REPLY: the words "Globally they took care of" have been replace "Overall they take care of"

Lines 134 to 136: did this retrospective study include some or all of the centres in the present survey?

REPLY All centers were included

Lines 146-147: This is somewhat of an over-statement about the frequency of Kingella kingae (and no reference provided). Unless this is a particular problem in Italy, it would be best to remove this sentence.

REPLY: OK the sentence has been deleted, according to your suggestion

Line 179: again change "molecules" to "antibiotics"

REPLY: the word "molecules" has been replace "antibiotics”

In general, this is a useful document for guiding future strategies in Italy and the graphics are particularly helpful. Suggestions as to the actual types of studies which should be performed would be helpful to investigators.

Reviewer 2 Report

The manuscript "Practical issues in early switching from intravenous to oral antibiotic therapy in children with uncomplicated acute haematogenous osteomyelitis: results from an Italian survey" by Elena Chiappini, et al. presents a survey that intends to compile the prevailing practice in Italian clinics to treat children with uncomplicated acute haematogenous osteomyelitis.

I believe it's a relevant topic and adheres to the theme of this journal and the authors have overall done a good job to filter the relevant clinics disseminating care for affected children. 

The authors presented data in a clear and lucid fashion using tables and percentages which easy to understand and interpret and contains valuable information. The number of patients (n) seems limited but given the rare nature of this infection its still a valuable resource for health care providers across Italy and otherwise. 

In summary, I believe the study is relevant and deserves publication.

Author Response

Thank you very much for your comments.